# MLFMF: Data Sets for Machine Learning for Mathematical Formalization

**Andrej Bauer**
Faculty of Mathematics and Physics
University of Ljubljana
Institute for Mathematics, Physics and Mechanics
Ljubljana, Slovenia
`andrej.bauer@fmf.uni-lj.si`

**Matej Petković**
Faculty of Mathematics and Physics
University of Ljubljana
Department of Knowledge Technologies
Jožef Stefan Institute
Ljubljana, Slovenia
`matej.petkovic@fmf.uni-lj.si`

**Ljupčo Todorovski**
Faculty of Mathematics and Physics
University of Ljubljana
Department of Knowledge Technologies
Jožef Stefan Institute
Ljubljana, Slovenia
`ljupco.todorovski@fmf.uni-lj.si`

## Abstract

We introduce *MLFMF*, a collection of data sets for benchmarking recommendation systems used to support formalization of mathematics with proof assistants. These systems help humans identify which previous entries (theorems, constructions, datatypes, and postulates) are relevant in proving a new theorem or carrying out a new construction. Each data set is derived from a library of formalized mathematics written in proof assistants Agda or Lean. The collection includes the largest Lean 4 library Mathlib, and some of the largest Agda libraries: the standard library, the library of univalent mathematics Agda-unimath, and the TypeTopology library. Each data set represents the corresponding library in two ways: as a heterogeneous network, and as a list of s-expressions representing the syntax trees of all the entries in the library. The network contains the (modular) structure of the library and the references between entries, while the s-expressions give complete and easily parsed information about every entry. We report baseline results using standard graph and word embeddings, tree ensembles, and instance-based learning algorithms. The MLFMF data sets provide solid benchmarking support for further investigation of the numerous machine learning approaches to formalized mathematics. The methodology used to extract the networks and the s-expressions readily applies to other libraries, and is applicable to other proof assistants. With more than $250\,000$ entries in total, this is currently the largest collection of formalized mathematical knowledge in machine learnable format.

## 1  Introduction

Applications of artificial intelligence to automation of mathematics have a long history, starting from early approaches based on a collection of hand-crafted heuristics for formalizing new mathematical concepts and conjectures related to them [Lenat, 1977]. In the last decade, there has been a growing interest in formalization of mathematics with *proof assistants*, which verify the formal correctness of

37th Conference on Neural Information Processing Systems (NeurIPS 2023) Track on Datasets and Benchmarks.

mathematical proofs and constructions, and help automate the tedious parts. The trend is correlated with the interest of machine learning community in aiding formalization efforts with its expertise.

Machine learning methods are often used to address *premise selection*, i.e., recommendation of theorems that are useful for proving a given statement. DeepMath [Irving et al., 2016] proposes using convolutional and recurrent neural networks to predict the relevance of a premise for proving the given statement. While many other approaches [Polu and Sutskever, 2020, Welleck et al., 2022] use transformers and general language models, Paliwal et al. [2020] have shown that taking into account the higher-order structure of logical expressions used in formalizing mathematics can greatly improve the performance of premise selection and automated proving. Indeed, many approaches use graph neural networks to learn from the higher-order structures, e.g., [Wang et al., 2017]. More recently, graph neural networks have also been proven useful for explorative, unsupervised approaches to automated theorem proving with reinforcement learning [Bansal et al., 2020, Lample et al., 2022]. Some of these approaches address alternative tasks, such as recommending or automatically selecting suitable *proof tactics*, i.e., routines for performing a series of proof steps, applying a decision procedure, or for carrying out proof search.

Data sets of different origins have been used to evaluate the proposed approaches. Welleck et al. [2022] evaluate their approach on a selection of three hundred proofs included in the ProofWiki [proofwiki] library of mathematical proofs written in a combination of natural language and LaTeX. Polu and Sutskever [2020] use a standard library of the Metamath proof assistant. Lample et al. [2022] combine proofs from the Metamath library with proofs from the Mathlib library [Mathlib] of the Lean proof assistant. The latter has also been used for evaluating the approaches in [Han et al., 2022]. Wang et al. [2017], Paliwal et al. [2020], Bansal et al. [2020] evaluate their models within the HOL Light proof assistant based on higher-order logic [Harrison, 2009]. The formalized proofs in standard HOL libraries have been transformed into a HOLStep data set for machine learning, where examples correspond to more than 2 million steps from $11\,400$ proofs [Kaliszyk et al., 2017]. The training set includes proof steps in context (local hypotheses and the current statement being proved) and the library entry used in the step. Descriptions of the library entries are included in human-readable and machine-readable, tokenized versions. The data set has been recently upgraded to the interactive benchmark environment HoList for training automated proof systems with reinforcement learning [Bansal et al., 2019].

We present a collection of data sets, MLFMF, based on libraries of formalized mathematics encoded in two proof assistants, Agda and Lean. It supports evaluation and benchmarking of machine learning approaches for recommendation systems in the context of formalized mathematics.

| ID | entry |
|---|---|
| $\mathbb{N}$ | $\mathbb{N}$ : 
    `zero`: $\mathbb{N}$ 
    `suc(n)`: $\mathbb{N} \to \mathbb{N}$ |
| $+$ | $0 + n = n$, for all $n \in \mathbb{N}$ 
 $\mathtt{suc}(m) + n = \mathtt{suc}(m + n)$, for all $m, n \in \mathbb{N}$ |
| Lemma 1 (L1) | $m + 0 = m$ for all $m \in \mathbb{N}$. 
 This is proved by induction on $m$. |
| Lemma 2 (L2) | $m + \mathtt{suc}(n) = \mathtt{suc}(m + n)$, for all $m, n \in \mathbb{N}$. 
 This is proved by induction on $m$. |
| Theorem (T) | $m + n = n + m$, for all $m, n \in \mathbb{N}$. 
 This is proved by induction on $m$. In the base case ($m = 0$), we need 
 L1. In the induction step ($m = \mathtt{suc}(\ell)$), we need L2. |

Table 1: An example formalization of proof that the addition of natural numbers is commutative.

We transform each library into a directed multi-graph whose nodes represent library entries (theorems, lemmas, axioms, and definitions), while edges represent the references between them. Consider the example in Table 1. It starts with a definition of the set of natural numbers with two simple constructors that define the first natural number 0 and constructs all the others inductively by asserting that a successor `suc(n)` of a natural number `n` is also a natural number. The definition of the addition of natural numbers follows their definition by asserting two simple rules for the left addition of 0 and the left addition of a successor. Note that the definition of $+$ references the definition of $\mathbb{N}$. Next, the first lemma establishes the rule for the right addition of zero as the first simple commutativity case.

The second lemma establishes the right addition of a successor as the second case. The theorem at the end references the two lemmas to show (and prove) the commutativity of adding natural numbers.

The entries from Table 1 are transformed into a multi-graph depicted in Figure 1a. It contains five nodes, each corresponding to a table row. The multi-graph includes an edge from the node $+$ to the node $\mathbb{N}$, indicating the reference to the set of natural numbers in the definition of addition. It also contains the self-reference of $+$, since the second case of this definition is recursive. Similarly, there are four edges from the theorem node to the two lemma nodes and the two nodes defining natural numbers and addition thereof. The obtained data allows us to approach premise selection as a standard edge prediction machine learning task.

Furthermore, we transform each formalized entry into a directed acyclic graph that retains complete information about the entry, see Figure 1b. By including the entire entry structures in the data sets, we make them suitable for further exploration of the utility of the state-of-the-art approaches to graph-based machine learning. A detailed description of the format is given in Sections 3.3 and 3.4.

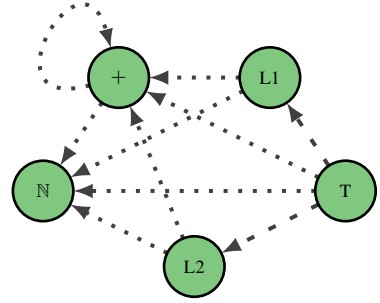

```
(:entry
  (:name ℕ)
  (:type (...))
  (:data
    (...)
    (:name ℕ.zero)
    (:name ℕ.suc)
  )
)
```

(a) The multi-graph representation of the example proof from Table 1.

(b) An s-expression from which we obtained the DAG for the entry $\mathbb{N}$ in Table 1.

Figure 1: The two-part representation of a library. Library as a whole is represented as a network of references (a). Additionally, every entry is represented as a DAG which is shown here in its textual s-expression format (b). Note that some nodes of DAG were replaced by (...) for better readability.

Our approach is general and can be applied to other proof assistants based on type theory. Moreover, even though Agda and Lean have quite different internal representations, the corresponding data sets use a common format that requires little or no knowledge about the inner workings of proof assistants. Thus our collection provides the machine learning community with easy access to a large amount of formalized mathematics in familiar formats that allow immediate application of machine learning algorithms. To our knowledge, MLFMF is the first and most extensive collection of data sets featuring more than one proof assistant and providing access to the higher-order structured representation of more than 250 000 mathematical formalization entries.

## 2  Formalized Mathematics

Formalized mathematics is mathematics written in a format that allows algorithmic checking of correctness of mathematical proofs and constructions. The programs that perform such checking are called *proof assistants* or *proof checkers*. An early proof checker was AUTOMATH [Bruijn, 1970], while today the most prominent assistants are Isabelle/HOL [Isabelle, HOL, Harrison], Coq [Coq], Agda [Agda] and Lean [de Moura et al., 2015]. They are all *interactive*: As the user develops a piece of formalized mathematics the assistant keeps track of unfinished proof goals, displays information about the current goal, checks the input on the fly, and provides search and automation facilities.

The level of automation varies between different proof assistants. In Agda, which supports little automation, the user directly writes down proofs and constructions in abridged type-theoretic syntax that Agda checks and algorithmically elaborates to fully formal constructions. On the other end of the spectrum are Isabelle/HOL and Lean, where the user relies heavily on *tactics*, which are routines that automatically perform various tasks, such as running a domain-specific decision procedure, applying a heuristic, or carrying out proof search.

The mathematical formalism most commonly used as the underpinning of a proof assistant is type theory, of which there are many variants [Church, 1940, Martin-Löf, 1975, Coquand and Huet, 1988]. The proof assistant processes the user input by disambiguating mathematical notations, applying tactics and other meta-level processing commands, and internally stores the resulting proofs, theorems, constructions, definitions, and types as expressions, or syntax trees, of the chosen type theory. These are typically quite verbose, so that checking their correctness is straightforward, but contain many more details than a user may wish to look at.

Libraries of formalized mathematics comprise units, organized hierarchically with a module system or namespaces, each of which contains a number of entries: definitions of types, constructions of elements of types, statements and proofs of theorems, unproved postulates (axioms), as well as meta-level content, such as embedded documentation, definitions of tactics, hints for heuristics, and other automation mechanisms.

In the last decade the libraries of formalized mathematics have grown considerably, most recently with the rise of the popularity of the Lean proof assistant and the Mathlib library [Mathlib, community, 2019], around which a mathematical community of several thousand mathematicians has formed. Such growth presents its own challenges, many of which are of the software engineering kind and can be so addressed. In our work we addressed the specific problem of *recommendation*: given a large body of formalized mathematical knowledge, how can the proof assistant competently recommend theorems or constructions that are likely useful in solving the current goal? There are two typical scenarios: the user knows which theorem they would like to use but have a hard time finding it in the library, or the user is not aware of the existence of a potentially useful theorem that is already available. Both are obvious targets for machine-learning methods.

## 3  MLFMF Data Sets

In this section we describe our data sets in detail. We first explain the semantic content of the data extracted from libraries of formalized mathematics, describe the format and information content of the data sets, continue by reviewing the machine learning tasks for which the data sets were built, and finish with an overview of the technical aspects of the library-to-data-set transformation process.

### 3.1  The Extracted Data

Formalized mathematics is written by the user in a domain-specific language, often called the *vernacular* or the *meta-language*. The proof assistant evaluates the source code, which involves executing tactics, decision procedures, etc., verifies that the proofs and constructions so generated are mathematically valid, and stores the results using an internal type-theoretic format. One may apply machine learning techniques directly on the vernacular, as written by the user, or on the formal representation of mathematics. The former approach roughly corresponds to learning how to *do* formalized mathematics, and the latter what formalized mathematics *is*.

We took the latter approach, namely learning on the formalized mathematics itself, for two reasons. First, because we aimed at a uniform approach that is applicable to most popular proof assistants, it made sense to use the internal type-theoretic representations, which are much more uniform across proof assistants than the vernaculars. Second, the vernacular contains meta-level information, such as what tactics to use, from which one cannot discern directly which theorems are actually used in a given proof. Without this information, one can hardly expect a recommendation system to work well.

Every data set that we prepared is generated from a library of formalized mathematics. Most libraries, and all that we incorporated, are organized hierarchically into modules and sub-modules, each of which is a unit of vernacular code that, once evaluated by the proof assistant, results in a list of *entries*: definitions of types, constructions of elements of types, theorems and their proofs, and unproved postulates. The entries refer to each other and across modules, possibly cyclically in case of mutually recursive definitions.

The internal representations of entries vary across assistants, but all have certain common features:

1. Each entry has a *qualified name* $M_1.M_2 \ldots M_k.N$ by which it is referred to, where $M_1.M_2 \ldots M_k$ is a reference to a module in the module hierarchy and $N$ is the local name of the entry, for example `Algebra.Group.FirstIsomorphismTheorem`.

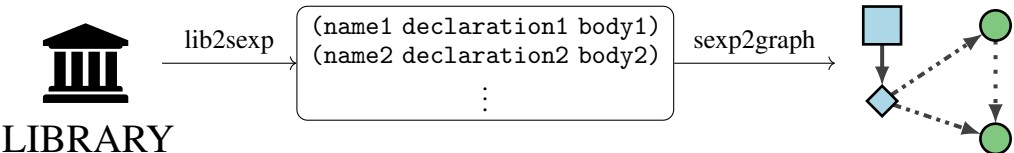

Figure 2: The two stages of the data transformation. First, a language-dependent (i.e., Agda or Lean) command line tool is used to transform the library entries into s-expressions. In the second stage, Python scripts are used to explicitly construct the directed multi-graph, which contains library modules, entries, and references among them.

2. Each entry has an associated *type T*, which specifies the information content of the body of the entry. For example, the type List($\mathbb{N}$) specifies that the entry is a list of natural numbers. Importantly, logical statements are just a special sort of types, so that the type of a proof is the logical statement that it proves. (This is to be contrasted with first-order logic, where logical statements are strictly separated from types.)

3. An entry has a *body*, which is an expression of the given entry type. In some cases the body may be missing, for instance if the user declares an axiom.

4. Depending on the proof assistant, various *meta-level information* is included, such as which arguments to functions are implicit (need not be provided by the user).

## 3.2 Data Description

In this section, we describe the data set. We start with a brief description of data transformation process, continue with the detailed description of the resulting pair of computational graphs for the entries in the library, and the directed, multi graph of references in the library (see 3.3) and 3.4).

Every data set consists of two parts. The first part is a set $\mathcal{T}$ of abstract syntax trees (AST) that correspond to the entries in the library, while the second is a directed multi-graph $G(V, E)$, where $V$ is a set of library entries, and $E$ includes the references among them. ASTs are actually trees in the case of Agda libraries. However, in Lean, they are directed acyclic graphs (DAGs) due to memory optimization and node-sharing: all the parents that would potentially reference their own copy of a node (or a subtree), rather reference the same node. For this reason, we refer to them as computational graphs. They provide the full information about every entry in the library and are given in the s-expression format that is much easier to parse, as compared to the typically very flexible syntax of proof assistants that allows for implicit arguments, mix-fix notation, etc. For example, the function if_then_else x y z in Agda can be called as if x then y else z. Learning from the source code would put an additional burden on the machine learning algorithm. Learning directly from computational graphs, on the other hand, is much easier.

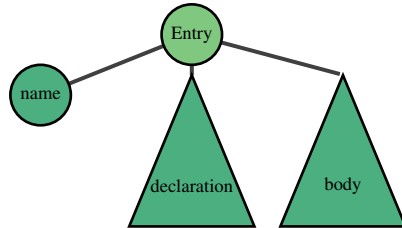

Figure 3: The DAG representing an entry has a single root node with three children: a node containing the entry name, a DAG containing the entry declaration, and a DAG representing the entry body.

## 3.3 The Computational Graphs

During compile time, the full type of every entry in the library is computed and the source code of the entry is converted to a (directed acyclic) computational graph. We intercept this procedure and export every entry as a Lisp *s-expression*, which is defined recursively as:

1. A literal is an s-expression, and

2. A list of s-expressions is an s-expression.

For example, the literals `12` and `"foo"` are s-expressions, and the list `("foo" ("bar" 12) "baz")` is also an s-expression with three elements: `"foo"`, `("bar" 12)` (which contains two s-expressions) and `"baz"`. Every s-expression that is obtained from the entries in a library is three-part, as shown in Fig. 3. In consists of the name of the entry, the s-expression that describes the declaration, and the s-expression that describes the body of the entry.

Even though the entries were manually encoded and mostly take at most a few kilobytes of space, their computational graphs can be much larger (more than a gigabyte), mostly due to the type checking and the expansion of the declared type of the entry.

### 3.4 The Multi-Graph

For simplicity reasons, we will refer to the directed multi-graph $G(V, E)$ simply as a graph. Its meta-structure is shown in Fig. 4a. In the description below, we follow this structure (in the bottom-up manner) and the concrete example of a subgraph for Agda's standard library in Fig. 4b.

**Entry nodes.** Every module in a library defines at least one entry (shown as green circles), e.g., `Bijection DEFINES id`, `Bijection DEFINES Bijection` (these are two different nodes), and `Injection DEFINES injective`. We further differentiate between different kinds of entries, as shown in Tab. 2. Most of the nodes in the graph are entries (and most of them are `functions`), and most of the edges are of type `REFERENCE FROM DECLARATION/BODY`.

**Library and module nodes.** The only nodes with no incoming edges (root nodes) are the library nodes (shown as blue squares). Every graph contains at least one library node—the one that corresponds to the library itself. In Fig. 4b, this is the node `stdlib`. However, the graph might contain

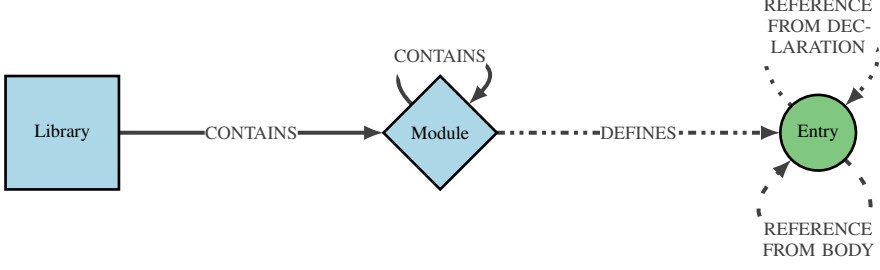

(a) The meta-structure of the graph.

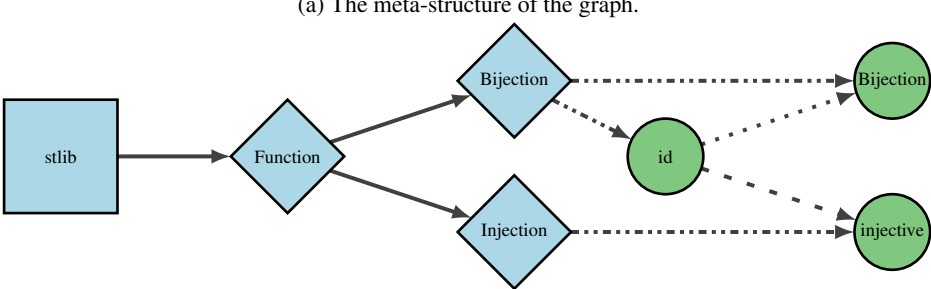

(b) An excerpt from Agda's standard library. The entry `id` (indentity function) is defined in the module `Bijection`, which is a submodule of the module `Function`, which is one of the top modules of the library. In the declaration of `id`, the entry `Bijection` is referenced. In the body of `id`, the entry `injective` is referenced. The two referenced entries are defined in the submodules `Bijection` and `Injection`, respectively.

Figure 4: A meta-graph of libraries (a) and a subgraph of the graph that was created from Agda's standard library (b) that follows the prescribed meta-structure.

Table 2: Tags of the nodes in s-expressions.

| kind | description |
|---|---|
| :data | inductive data type (natural numbers, lists and trees) |
| :constructor | data-type constructor (successor, cons) |
| :function | function (including constants as nullary functions) |
| :record | record type (a structure with named fields or attributes) |
| :axiom | postulated type or statement (no inhabitant or proof given) |
| :primitive | built-in (primitive) function |
| :sort | the sort of a type (proposition, universe at a given level) |
| :recursor | the induction/recursion principle associated with an inductive data-type |
| :abstract | entry whose body is hidden |

an additional node `outer library` if any of the library entries reference some external entries that are not part of the library (for example, built-in types). The library nodes are directly connected to the nodes representing modules (shown as blue diamonds) via the edges of the type CONTAINS, e.g., `stdlib CONTAINS Function`. Every module can contain zero or more (sub)modules, e.g., `Function CONTAINS Bijection`.

In the case of Agda libraries, the module nodes correspond to the modules that are actually present in the library and resemble the file system of the library. Lean, however, supports the use of namespaces. If the file `a/b/c.lean` defines an entry `foo.bar.F`, and the file `d/e.lean` defines an entry `foo.bar.G`, those two entries are part of the same namespace `foo.bar` and the exact location in the file system where these two entries were defined, is irrelevant. Therefore, module nodes for Lean's library Mathlib4 correspond to namespaces in the library. Following the previous example, we create module nodes `foo` and `bar`, together with the edge `foo CONTAINS bar`.

### 3.5 Machine Learning Tasks

The main motivation for the creation of the data set was the development of machine learning algorithms that would enhance current proof assistants and help mathematicians using them. This translates to the following two machine learning tasks.

**Link prediction.** Given the current state of the multi-graph of references among the entries, learn a model that predicts the future, novel links (references) among the library entries. Formally, we learn a model $M : (u, v) \mapsto M(u, v) \in [0, 1]$ that given two nodes $u$ and $v$ outputs the model confidence in the presence of the edge $(u, v)$. The (current) computational graphs of the entries can be used as additional information for learning such a model. If learning from the multi-graph only, one can use standard node- or edge-embedding approaches as well as graph neural networks.

**Recommendation.** The problem of predicting the future references among the entries could be understood as a recommendation task as well. Given a specific unfinished entry (possibly with some additional context, such as the list of lemmas/ claims that were used last), the task is to recommend the candidates that could be referenced in the current computational graph of the entry to complete it.

Note that the two tasks are equivalent, i.e., solving one solves the other. A link prediction model $M$ (see above) can be converted into a recommendation system by fixing the entry $u$ and recommending the entries $v \in V$ with the highest confidence levels $M(u, v)$. Vice versa, given a recommendation model $M' : u \mapsto M'(u) \subseteq V$, we can define a corresponding link-prediction model $M$ as $M(u, v) = 1$ if $v \in M'(u)$, and $M(u, v) = 0$ otherwise.

Since the essential part of the MLFMF data set is a directed multi-graph (which represents a heterogeneous network), other standard learning tasks for graphs/ networks might also be interesting. Here, we mention two example instances of the common node classification task.

**Entry class detection.** A straight-forward instance of node classification task would be classifying the entries into their types from Table 2, e.g., `function` or `axiom`. This does not require additional manual labeling and should not be too hard, especially when computational graphs are taken into account, since the structure of a `function` is quite different from the structure of, e.g., `record`.

**Claim detection.** A more challenging instance of node classification is predicting whether a `function` entry is a claim (e.g., a lemma, corollary, theorem, etc.) or not, since some of the entries are simply definitions of, for example, the addition of natural numbers. Approaching this task, however, would require additional (manual) labeling of the entries.

### 3.6 License

We make MLFMF publicly available under the Creative Commons Attribution 4.0 International[1] (CC BY 4.0) license at `https://github.com/ul-fmf/mlfmf-data`.

## 4 Experiments and Results

In this section, we first introduce the experimental setup for the baseline experiments (how to prepare the train and test part of the data set, and which standard metrics can be used), and then, after briefly introducing the baseline methods, we report the experimental results.

### 4.1 Train-test split

When splitting the graph into train and test data sets, we should split the multi-graph $G(V, E)$ and the set of computational graphs of the entries. In the case of the link prediction and recommendation tasks, we should focus on `function` nodes, since these are the only nodes that correspond to a computational graph whose body contains a proof of a claim formalized in the declaration part of the computational graph.

In our baseline experiments from Sec. 4.4, we follow a generic approach to creating a train-test split. The approach takes two parameters: $p_{\text{test}} \in (0, 1)$, $p_{\text{body}} \in [0, 1)$. First, we randomly choose the proportion $p_{\text{test}}$ of `function` nodes. We assume that those correspond to partially written entries whose computational graphs have completely specified type, i.e., the user knew how to formalize a claim, and *partially* known body, i.e., the proof of the claim is not finished yet. Note that, often, proofs are not written linearly and might contain so-called holes at problematic parts where the right lemmas are yet to be applied (possibly with already known arguments). Thus, we need to modify the computational graphs of the test nodes to reflect the changes in the multi-graph.

We simulate the applications of the missing lemmas by keeping only the proportion of $p_{\text{body}}$ of the references in the body. Since our graph contains a weighted edge `u REFERENCE FROM BODY v`, which we either remove or keep intact, we remove all references to $v$ from the body of $u$ or none of them. Then, the unfinished proofs are simulated by keeping the proportion of $p_{\text{body}}$ of the body of $u$, which is done by iterative pruning of the leaves of the body. At each iteration, a leaf is chosen uniformly at random. If the chosen leaf is a reference that we have to keep, the leaf is not pruned and we continue with the next iteration.

The removed edges represent positive test examples, and the negative test examples for learning predictive models need to be sampled. In the baseline experiments, the negative test examples were sampled uniformly at random.

### 4.2 Evaluation metrics

For link prediction, one can use standard classification metrics, such as accuracy, precision, recall, and $F_1$-score. If the model returns its confidence $M(u, v) \in [0, 1]$ instead of the class value ($M(u, v) \in \{0, 1\}$), one could additionally consider area under the receiver-operating-characteristic curve. Similarly goes for the recommendation models: one can use precision and recall.

If the recommendation model returns the relevance score of a candidate entry to the current context, we can rank candidates according to the score values, with the top recommendation having a rank of one (1). We can then compute the minimal (and the mean) rank of the actual references and average them over the testing examples. This is an important metric, since it counts the number of false recommendations with better ranks than any of the actual references. Ideally, the minimal rank is close to one, i.e, the top-ranked recommendation mostly matches the missing entry to be referenced.

---

[1]https://creativecommons.org/licenses/by/4.0/

### 4.3 Baseline Methods

**Dummy recommender.** This recommender ignores the current context and always recommends the $k$ nodes of the multi-graph with the highest in-degree.

**Bag of Words/TFIDF recommenders.** Bag of Words (BoW) recommender converts every computational graph $g(u)$ of an entry $u$ in a library into a bag of words $\mathrm{BoW}(u)$. We compute the relevance $M(u,v)$ of the candidate entry $v$ for the current context $u$ using the Jaccard similarity between the corresponding bag-of-words:

$$J(\mathrm{BoW}(u), \mathrm{BoW}(v)) = \frac{|\,\mathrm{BoW}(u) \cap \mathrm{BoW}(v)|}{|\,\mathrm{BoW}(u) \cup \mathrm{BoW}(v)|}.$$

Similarly, TFIDF-recommender embeds $g(u)$ into a term-frequency-inverse-document-frequency vectors (obtained from the corresponding bags-of-words) as implemented in Scikit-Learn 1.2.2 [Pedregosa et al., 2011]. The relevance of the candidate entry $v$ is computed as a Manhattan or a cosine distance between the TFIDF-vectors of $u$ and $v$.

**FastText embedding recommender.** It embeds every computational graph $g(u)$ into a vector $\vec{\varphi}(g(u)) = \sum_{\mathrm{word} \in g(u)} w(\mathrm{word}, g(u)) \cdot \varphi_{\mathrm{cc}}(\mathrm{word})$, where $\varphi_{\mathrm{cc}}(\mathrm{word})$ is the vector of `word` obtained from the fastText model trained of Common Crawl [Mikolov et al., 2018], and $w(\mathrm{word}, g(u))$ is the TFIDF weight of the word in the entry $u$.

**Recommendations via analogies.** We design a recommender that is based on fastText analogies property, i.e., the fact that $x = $ queen is one of the approximate solutions of $\varphi_{\mathrm{cc}}(\mathrm{king}) - \varphi_{\mathrm{cc}}(x) = \varphi_{\mathrm{cc}}(\mathrm{man}) - \varphi_{\mathrm{cc}}(\mathrm{woman})$. We design the *analogy recommender* that for a given entry $u$ recommends the nodes $v$, for which a good analogy $u' \to v'$ of the edge $u \to v$ can be found. The relevance of the candidate entry $v$ in a given context $u$ is defined in terms of the Manhattan distance as

$$r(u,v) = 1 \quad / \min_{u' \to v' \in E(G)} \|[\varphi_{\mathrm{cc}}(u) - \varphi_{\mathrm{cc}}(v)] - [\varphi_{\mathrm{cc}}(u') - \varphi_{\mathrm{cc}}(v')]\|_1.$$

**Node2vec-based link prediction.** We train a node2vec [Grover and Leskovec, 2016] model (as implemented in Gensim 4.3.1 [Rehurek and Sojka, 2011] on the multi-graph to obtain node embeddings. We obtain the embedding of the edge $(u,v)$ by concatenation of the node embeddings for $u$ and $v$. A tree-bagging classifier $M : \varphi(u \to v) \mapsto M(\varphi(u \to v)) \in [0,1]$ is trained on the tabular data obtained with using the edge embeddings as inputs and the edge presence as the target to be predicted. We selected bag of trees ensemble since it is a robust classifier, working well on tabular data, when using the recommended settings of 100 fully grown (not pruned) classification trees.

We selected baseline methods that are not computationally expensive and are robust to hyperparameter settings: if not mentioned otherwise, the methods use the default parameter settings. We can combine multiple embeddings (e.g., those from node2vec together with those from fastText) as the input to the similarity measure of the recommender or classifier for the link prediction task. However, as noted in the next section, this did not improve the best results. In all the experiments, we generated the train-test split with the parameters $p_{\mathrm{test}} = 0.2$ and $p_{\mathrm{body}} = 0.1$. The results here are reported for $k = 5$ recommended items and the threshold $\vartheta = 0.5$ for classification.

### 4.4 Results

The experiments on Lean were run on a computer with 4 Intel Core i7-6700K CPU cores and 64 GB of RAM. The experiments on Agda were run on a smaller machine (2 Intel Core i7-5600U CPU cores, 12 GB of RAM). Experiments that would last more than a week were not carried out (analogies on the Type Topology and Mathlib4 libraries, and fastText on the Mathlib4 library).

Tab. 3 reports the results of the experiments: for extended report including other evaluation metrics (accuracy@k, area under the ROC curve, etc.), and ablation study of node2vec on Agda stldib, check the supplementary material. For the algorithms that were run with more than one parameter setting, the best results are reported (for example, TFIDF was run with cosine- and Manhattan-based similarity measures). The best-performing baseline method is node2vec. It is the only one that ranks on average at least one actual reference among the ten most relevant candidate references for the three

Agda libraries. However, it fails to do so for Lean Mathlib4 and this can be only partially explained by the size of the Mathlib4. Note that node2vec is also the only one that explicitly learns from the multi-graph and, apparently, humans writing proofs in Agda, structured the references better than the computers in Lean, where built-in search heuristics (tactics) are used. The multi-graph is partially used by the analogies recommender as well, since the candidate recommendations are evaluated by considering the existing references $u' \to v'$ in the library. This might be the reason for its good performance on the Agda stdlib data set.

Table 3: The accuracy (acc) and minimal rank of the true reference for the MLFMF data sets. The best results (bold) are obtained with a combination of a node2vec and a tree-bagging classifier.

| method | Agda stdlib | | Agda unimath | | Agda TypeTopology | | Lean Mathlib4 | |
|---|---|---|---|---|---|---|---|---|
| | acc | minRank | acc | minRank | acc | minRank | acc | minRank |
| Dummy | 0.51 | 218 | 0.53 | 2134 | 0.50 | 4556 | 0.51 | 26065 |
| BoW | 0.50 | 1608 | 0.50 | 1571 | 0.50 | 4496 | 0.50 | 15458 |
| TFIDF | 0.51 | 144 | 0.52 | 112 | 0.51 | 552 | 0.51 | 443 |
| fastText | 0.51 | 132 | 0.52 | 394 | 0.50 | 1292 | NA | NA |
| analogies | 0.52 | 37 | 0.51 | 158 | NA | NA | NA | NA |
| node2vec | **0.96** | **4.37** | **0.96** | **3.24** | **0.98** | **5.81** | **0.95** | **195** |

Surprisingly, TFIDF embeddings perform no worse (or even better) than FastText embeddings. The reason for this might be that many words, such as group, ring, etc. have different meanings in mathematics than in general texts. Note that we tried to run additional experiments with the combination of node2vec and TFIDF/fastText embeddings, but accuracy and minRank were both worse, as compared to the node2vec results.

In sum, the baseline results show that the information on the structure of the multi-graph is crucial for obtaining classifiers with performance beyond the default performance of the dummy baseline. The recommendation performance, measured as mean minimal rank, is valuable enough (less than five recommendations to be checked to find the right one) for two Agda libraries. Developing sound recommendation systems for the other two libraries remains a challenge to be addressed by machine learning methods beyond the baselines considered here.

## 5 Conclusion

We introduced MLFMF, a suite of four data sets corresponding to three libraries in Agda and one library in Lean proof assistants. It includes almost $250\,000$ entries, i.e., definitions, axioms, and theorems with accompanying proofs. References between entries are included in a multi-graph, where nodes are entries, edges represent references among the entries, and each entry is represented with a direct acyclic graph reflecting the structure of the entry source code encoding. Such a structure provides machine learning researchers with an opportunity to address the task of recommending relevant entries for the goal at hand as a standard edge prediction task. Such a representation of the entries allows for use of graph-based methods that can exploit the structural and semantic information stored in the multi-graphs. The report on the results of the baseline methods establishes a benchmark for comparative evaluation of future developments of machine learning for mathematical formalization that goes beyond a single proof assistant.

A notable limitation of our data sets is the lack of information on the developmental evaluation of the libraries. If the latter could be followed, the more realistic test nodes $V_{\text{test}}$ could be defined as the *latest* $|V_{\text{test}}|$ nodes encoded in the library. However, even with version control information (available since the libraries are stored in GitHub repositories but currently not included), determining the entries' chronological order might be computationally expensive. An approximation of the chronological order might be obtained by computing a topological ordering on the `function` nodes and selecting nodes from the tail of the ordered list. However, existing definitions in a library might be rewritten, so the accuracy of such an approximation is questionable.

Finally, we plan to include other, more recent libraries in our data set collection. The newly incorporated libraries might include references to earlier, standard libraries, providing further opportunities for real-world testing scenarios.

## 6 Acknowledgments

This material is based upon work supported by the Air Force Office of Scientific Research under award number FA9550-21-1-0024. The authors also acknowledge the financial support of the Slovenian Research Agency via the research core funding No. P2-0103 and No. P1-0294.

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
