# MLFMF: Data Sets for Machine Learning for Mathematical Formalization Supplementary Material

**Andrej Bauer**
Faculty of Mathematics and Physics
University of Ljubljana
Institute for Mathematics, Physics and Mechanics
Ljubljana, Slovenia
`andrej.bauer@fmf.uni-lj.si`

**Matej Petković**
Faculty of Mathematics and Physics
University of Ljubljana
Department of Knowledge Technologies
Jožef Stefan Institute
Ljubljana, Slovenia
`matej.petkovic@fmf.uni-lj.si`

**Ljupčo Todorovski**
Faculty of Mathematics and Physics
University of Ljubljana
Department of Knowledge Technologies
Jožef Stefan Institute
Ljubljana, Slovenia
`ljupco.todorovski@fmf.uni-lj.si`

This document provides several pieces of meta-information about the MLFMF data set collection, as well as some additional details and results from the experiments.

All the code is available at the GitHub repository `https://github.com/ul-fmf/mlfmf-data`. For a detailed description of the preprocessing scripts and the script for running the model, please refer to the README in the repository. However, due to space limitations, all the preprocessed data can be found at `https://doi.org/10.5281/zenodo.10041075`, which – for self-sufficiency reasons – contains the relevant preprocessing tools and a simple script to load the data sets.

## 1  Data Collection

We obtained the source code of the libraries from their publicly available GitHub repositories. At the time of collection, we retrieved the latest versions of the libraries, which are specified in Table 1. In the case of Agda, a fork (`https://github.com/andrejbauer/agda/tree/master-sexp`) of the official Agda repository (`https://github.com/agda/agda`) was created to modify the compilation procedure, so that it outputs s-expressions (see the main article, Section 3). In the case of Lean, a standalone tool (`https://github.com/andrejbauer/lean2sexp`) was developed and implemented in Lean to process entries in Lean's `olean` binary format.

Table 1: The versions of the four libraries transformed to the data sets.

| GitHub repository | commit |
|---|---|
| agda/agda-stdlib | cfa2504316d7e03e4a1a5f2353976796e07a9f1e |
| UniMath/agda-unimath | bca22ae30b07f9ee02f82d1f4893ddc389185cb0 |
| martinescardo/TypeTopology | 8b44abc0c99775c8141709a47a52ec5827357886 |
| leanprover-community/mathlib4 | cdbd878af82f017d6a1db38e5581a348d7706002 |

A co-author of this paper developed both conversion procedures, and their anonymized implementations (without the links to the specific GitHub repositories) are present in the Google Drive folder

mentioned above. We plan to make them publicly available upon acceptance and/ or public release of the data set.

## 2 Data Description

As described in the article manuscript (see Section 3.2), every data set corresponds to a library of formalized mathematics. The data set consists of two parts: a set of computational graphs of the library entries (described in Section 3.3 of the article) and a directed multi-graph (see Section 3.4). We store every computational graph in a separate `.dag` file (a text file whose extension `dag` stands for directed acyclic graph). The second part of the data set, the directed multi-graph, or network for short, is stored in a text file, listing its nodes and links.

In the following subsections, we describe the file format and the structure of the repository.

### 2.1 The Structure of the DAG Files

Every DAG file contains a tab-separated table with four columns of `NODE ID`, `NODE TYPE`, `NODE DESCRIPTION` and `CHILDREN IDS`, described in Table 2. Excluding the header, every line in the file represents a single DAG node. For example, the row

```
2923823 :name "Relation.Binary.Definitions.Transitive 70" []
```

describes a node with the ID 2923823, which is a `:name` that refers to the fully qualified name[1] `"Relation.Binary.Definitions.Transitive 70"`, and has no children (empty list `[]`).

Table 2: The meaning of the four columns in the `.dag` files.

| column name | description |
| --- | --- |
| NODE ID | A unique identifier (integer) of a node in the corresponding computational graph. The uniqueness is guaranteed (and meaningful) locally within a single DAG file. The same `NODE ID` in different files does not denote that the corresponding nodes are the same or related. |
| NODE TYPE | A type of the node revealing its specific role in the computational graph, e.g., `:entry` declares a start of the new entry, `:name` specifies the name of the entry or its reference. |
| NODE DESCRIPTION | Additional information about the node. This might be the name that a `:name` node introduces or the value of the literal in the `:literal` nodes. For most of the node types, it is empty. |
| CHILDREN IDS | A list of unique identifiers of the children of the current node in the computational graph. |

Even though the data were prepared uniformly (as much as possible) for both proof assistants, Agda and Lean, only 20 node types are relevant for both Agda and Lean. Additional 45 are Agda-specific node types, while 19 are Lean-specific. Note that we included the node type for completeness and lossless transfer of the information available in the libraries. For many machine learning applications, the information on the node type can be ignored. In our experiments with the baseline methods (Section 4.3 of the article), we did not use the information on the node type. However, machine learning experts familiar with the detailed semantics of the definition types in the programming languages Agda and Lean can use the information about the node types.

### 2.2 The Structure of the Network Files

The network file lists the nodes and the links of the multi-graph of references among the entries and modules of the corresponding library. Lines starting with the word `node` represent nodes with two tab-separated fields

---

[1]In Agda there might be name duplicates, so an additional (Agda-provided) id of the entry was included into the name (70 in the example above).

Table 3: The four types of the links in the network.

| link type | property | description |
|---|---|---|
| DEFINES | none | A link from module to entry nodes denoting that the corresponding module defines the corresponding entry. |
| CONTAINS | none | A link from a library/ module node that contains another (sub-)module node. |
| REFERENCE_TYPE | w | A link $a \rightarrow b$ between two entry nodes $a$ and $b$ denoting that the entry corresponding to $a$ references the entry $b$ from the type (declaration) part of its computational graph $g(a)$. The property w is the count of the references from the type part of $g(a)$ to entry $b$. |
| REFERENCE_BODY | w | An analogue of REFERENCE_TYPE denoting a reference from the body part of the computational graph $g(a)$ to the entry $b$. |

```
node <node name> <node properties>.
```

The entry names are unique and include the modules (Agda) and namespaces (Lean) in the context in which they are defined. The only property of a node is its `label`, whose value is either one of the nine entry types given in Table 1 of the article, a `:module` or a `:library`. The latter two denote nodes corresponding to the library modules (in Agda) and the current library or the referenced libraries.

The lines starting with the word `link` are of form

```
link <source node> <sink node> <link type> <link properties>.
```

The first two fields refer to the source's and sink's node names. The following field is the link type, i.e., one of the types from Table 3, which also explains the link properties (the last field in the line). In addition to these link types, there are four Agda-specific link types: REFERENCE_<TYPE/BODY>_TO_<WITH/REWRITE>. These encode the references from actual user-defined entries (either from their types or bodies) to the entries created by the Agda compiler. For example, the definition

```
isEven : Nat -> Bool
isEven n with (mod2 n)
...            | 0      = true
...            | 1      = false
```

(which reads as *return true if $n \bmod 2 = 0$, and false if $n \bmod 2 = 1$*) contains a `with` block. This block is internally represented as a separate entry, referenced from the body of the entry `isEven`. These reference links have the same properties as the standard reference links from Table 3.

**External entries.** A given entry in a library might reference some built-in method of a proof assistant (as a programming language), e.g., a method on lists in Lean. In that case, the s-expression of the referenced method might not be available, but its fully qualified name is. We include the corresponding node and the corresponding external modules in the network. The only difference from the standard case is that we set the NODE TYPE of such modules as `:external-module`.

## 2.3 Data Size

Table 4 gives the basic size-related statistics of the datasets. We can see that Lean Mathlib4 is approximately ten times bigger than any of Agda's libraries. Next, most of the nodes in the network are entry nodes. The difference between the number of nodes and entries is due to the module nodes and the references to the nodes not being part of the library we processed. This often happens in Mathlib4 and sometimes in `stdlib`. The other two libraries (Unimath and TypeTopology) are—in that sense—self-contained.

Table 4: The number of entries, the total and maximal size of their compute graphs (nodes that they contain) and the size of the network $G(V, E)$.

| library | entries | total entry size | max entry size | $|V|$ | $|E|$ |
|---|---|---|---|---|---|
| Agda stdlib | 16,483 | $1.3 \cdot 10^7$ | $8.1 \cdot 10^3$ | 16,855 | 242,484 |
| Agda Unimath | 20,163 | $1.9 \cdot 10^7$ | $9.8 \cdot 10^5$ | 21,493 | 322,446 |
| Agda Type Topology | 31,232 | $3.6 \cdot 10^7$ | $3.3 \cdot 10^5$ | 31,701 | 726,710 |
| Lean Mathlib4 | 202,769 | $5.0 \cdot 10^8$ | $8.3 \cdot 10^6$ | 215,229 | 7,378,824 |

## 2.4 Additional details on the experiments

Here, we first give some additional details on the experiments in which node2vec was used. Then, we show an extension of the Table 3 with additional quality measures.

### 2.4.1 The detailed node2vec experimental setup

**Network preprocessing details.** Our network is a weighted directed multi-graph, while node2vec was designed for simple (possibly directed) graphs. We carry out the conversion to the undirected weighted graph in the following steps.

1. We transform the weights $w = w(u, v)$ on the directed edges $(u, v)$ via the TFIDF-like transformation. First, we create a document for every node $u$. The ID of every node $v$, such that $(u, v)$ is a directed edge, appears (as a single *word*) in the document $w(u, v)$ times (note that the weights are the counts of references). Second, the updated weight $w'(u, v)$ is the TFIDF-score of the (ID of) $v$ in the document that corresponds to the node $u$.

2. There can be more than one directed edge from $u$ to $v$. The weights on such edges from the previous step are merged into a single value by summation.

3. To obtain an undirected graph, the final weight on the (undirected) edge between $u$ and $v$ is the sum of the weights on the directed edges $(u, v)$ and $(v, u)$ (if the edges exist).

This transformation prevents walks from visiting *hubs* (nodes with extremely large degrees) too often and, at the same time, still takes the weights into account.

**Node2vec implementation details.** For Agda libraries, we were able to use the freely available implementation of node2vec (`https://github.com/eliorc/node2vec`), which precomputes the transition probabilities, generates the walks, and then uses Gensim's word2vec model. However, computing the transition probabilities for a graph $G(V, E)$ takes $\mathcal{O}(|E|^2)$ space, which was infeasible in the case of Lean Mathlib4. Therefore, we used our own implementation of walk generation (written in Python and compiled *just it time* with numba).

### 2.4.2 Extended results

Here, we first extend Table 3 from the main text (the main table with results) with additional evaluation measures. In addition to that, we show the results of an ablation study of node2vec on Agda standard library.

**Additional evaluation measures.** The additional performance measures are accuracy@k where $k = 5$ (Table 5), precision (Table 6), recall (Table 7), and area under the ROC curve (Table 8). Accuracy@k is a recommender system evaluation measure that gives us the average proportion of the correct recommendations in the top-$k$ recommendations. The rest of the measures are well-known evaluation measures for classification models. AU-ROC is threshold independent, whereas the precision and recall are reported at threshold $\vartheta = 0.5$.

Node2vec achieves the best accuracy@k for the three Agda libraries, while on the Lean Mathlib4 library, the dummy classifier (still) performs better. A similar situation can be observed with precision values, however, most of them should be considered trivial since the corresponding recalls (and the AU-ROC values as well) are rather low.

It turns out that the models are for most of the test edges $(u, v)$ quite sure, whether this edge should be present or absent, even though they are far from being perfect. For example, AU-ROC of node2vec

is 0.99 on Mathlib4, but its accuracy@k is (approximately) 0.00. This means that randomly sampling negative edges did not lead to extremely hard negative examples.

Table 5: The accuracy@k of the models on the considered data sets. The best results (bold) were achieved by Dummy and node2vec models.

| acc@k | Agda stdlib | Agda unimath | Agda TypeTopology | Lean Mathlib4 |
|---|---|---|---|---|
| Dummy | 0.16 | 0.10 | 0.12 | **0.09** |
| BoW | 0.00 | 0.02 | 0.00 | 0.00 |
| TFIDF | 0.04 | 0.06 | 0.03 | 0.05 |
| fastText | 0.00 | 0.05 | 0.02 | NA |
| analogies | 0.04 | 0.03 | NA | NA |
| node2vec | **0.29** | **0.27** | **0.15** | 0.00 |

Table 6: The precision (at $\vartheta = 0.5$) of the models on the considered data sets. The best results (bold) are obtained by various models, but only node2vec models have non-trivial recall as well.

| precision | Agda stdlib | Agda unimath | Agda TypeTopology | Lean Mathlib4 |
|---|---|---|---|---|
| Dummy | 0.98 | **1.00** | **1.00** | **1.00** |
| BoW | 0.94 | 0.97 | 0.93 | 0.98 |
| TFIDF | 0.98 | 0.99 | 0.99 | **1.00** |
| fastText | 0.98 | 0.98 | 0.98 | NA |
| analogies | **0.99** | 0.99 | NA | NA |
| node2vec | 0.97 | 0.97 | 0.98 | 0.94 |

Table 7: The recall (at $\vartheta = 0.5$) of the models on the considered data sets. The best results (bold) are obtained by node2vec models.

| recall | Agda stdlib | Agda unimath | Agda TypeTopology | Lean Mathlib4 |
|---|---|---|---|---|
| Dummy | 0.01 | 0.07 | 0.06 | 0.02 |
| BoW | 0.00 | 0.01 | 0.00 | 0.00 |
| TFIDF | 0.03 | 0.04 | 0.02 | 0.01 |
| fastText | 0.02 | 0.03 | 0.01 | NA |
| analogies | 0.03 | 0.02 | NA | NA |
| node2vec | **0.95** | **0.94** | **0.98** | **0.97** |

Table 8: The area under the ROC curve of the models on the considered data sets. The best results (bold) were achieved by node2vec models.

| AU-ROC | Agda stdlib | Agda unimath | Agda TypeTopology | Lean Mathlib4 |
|---|---|---|---|---|
| Dummy | 0.51 | 0.54 | 0.53 | 0.51 |
| BoW | 0.50 | 0.51 | 0.501 | 0.50 |
| TFIDF | 0.51 | 0.52 | 0.51 | 0.51 |
| fastText | 0.51 | 0.52 | 0.50 | NA |
| analogies | 0.51 | 0.51 | NA | NA |
| node2vec | **0.99** | **0.98** | **1.00** | **0.99** |

**Ablation study.** We performed an ablation study of node2vec on the Agda standard library in the following way. We keep the test set intact to obtain comparable results and only manipulate the training set by keeping the proportion $p \in \{0.1, 0.2, \ldots, 0.9, 1.0\}$ of the total weight of the edges in it. Since node2vec works in a transductive setting, all versions of the training set include all the nodes. To better understand the (aggregated) results, we show the distribution of minimal ranks for every training set as box plots in Figure 1. In addition to the results of node2vec on manipulated training sets, the figure also contains the results of the Dummy model on the original training set.

We can observe a large variance in the minimal ranks produced by the Dummy model. This is explained by the fact that the Dummy model always assigns top ranks to the most referenced entries

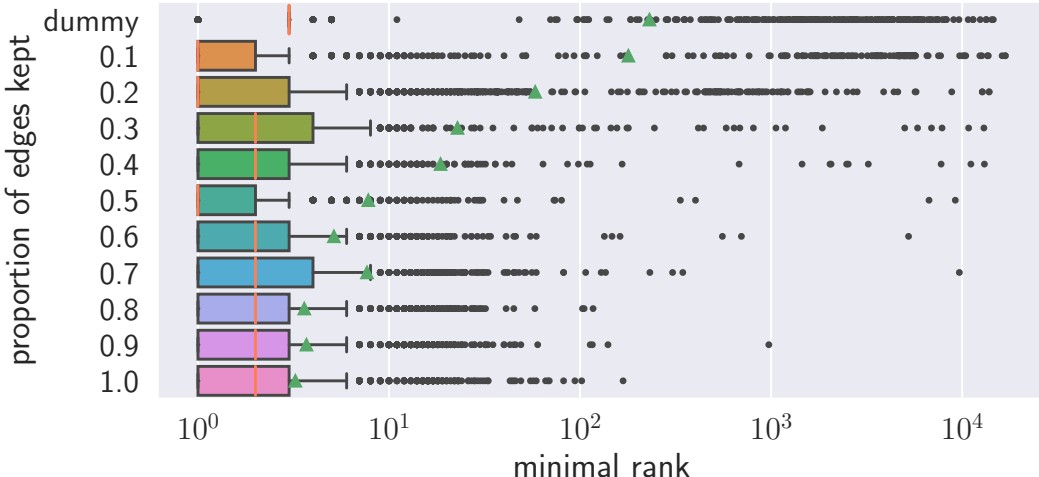

Figure 1: The distributions of the minimal ranks of the correct references for the test examples when varying the proportion of edges from the Agda stdlib library included in the training set from 0.1 and 1.0 (all edges). The box plot on the top corresponds to the Dummy recommendation system. The orange lines and the green triangles depict median and average minimal ranks, respectively.

while the ranks of the other entries are randomly distributed. Therefore, the Dummy model performs very well on the test entries that reference the most referenced entries, while at the same time, it performs pretty badly on the other test entries.

Analyzing node2vec results, we can see that training the model on only $p = 10\%$ of the edges already leads to a better performance of node2vec as compared to the Dummy model (in terms of the average minRank). As the value of $p$ increases from 0.1 to 1.0, the median of the minRank does not change much, while the average minRank is getting lower since there are fewer test examples on which the model performs *extremely* badly.

## 2.5   Repository and GDrive folder structure

The structure of the GitHub repository (preprocessing scripts and the code for learning) is described in its README file. In this document, we only describe the structure of the GDrive folder.

In the folder, every data set (library) resides in a separate directory with a name resembling the library name (e.g., `stdlib`). Each directory contains

- the file `network.csv`,

- a (zipped) directory `entries` with the `.dag` files.

The names of DAG files are two-part: the first part is the namespace to which a given entry belongs. This is the actual namespace of a Lean entry and the fully qualified module name of an Agda entry. The second part of the name is the entry number, which ranges from 0 (included) to the number of entries in an s-expression (excluded). For example, one of the `stdlib` entries can be found in `stdlib/entries/Data.Fin.Properties_0147.dag`.

All the data can be loaded by running `main.py` script. It needs two packages to run: `networkx` for storing the network and `tqdm` for showing the progress. They can be installed by issuing the command `pip install -r requirements.txt` (the requirements file is also present). The computational graphs are stored as members of tree-like class `Entry`, defined in `main.py`. The code was tested with Python 3.11.

For convenience, we uploaded `entries.zip` to the temporary Google Drive location. When running `main.py` for the first time, the directory `entries` will be automatically created and populated with unzipping the `entries.zip` file.

## 3   Intended uses

*MLFMF* data sets were created to support further improvement of the numerous machine learning approaches to formalized mathematics. Primarily, the data sets can be used to evaluate the efficiency of the recommendation systems used to support formalization of mathematics with proof assistants. These systems help humans identify which previous entries (theorems, constructions, datatypes, and postulates) are relevant in proving a new theorem or carrying out a new construction. Please refer to Sections 3.5 and Section 4 of the main article for further details.

However, the data set collection can also serve as an appropriate benchmark for machine learning from graphs. For example, the node types can be used as the target concept of node classification.

## 4   Hosting, Maintenance and Access

The data is available at `https://doi.org/10.5281/zenodo.10041075`. The link from the main text (`https://github.com/ul-fmf/mlfmf-data`) points to a repository that contains the rest of the code, as well as the README file.

After the reviewing process, the collection of the data sets will be published as is in the aforementioned (publicly available) GitHub repository under the Creative Commons Attribution 4.0 International[2] (CC BY 4.0).

Note that the data sets are based on the source code of Agda and Lean libraries that evolves through time (entries might get added, deleted, or modified), but we will obtain a persistent dereferenceable identifier for the current snapshot. Moreover, we plan to update each data set when the underlying library significantly changes. This is not a rare event, given that, e.g., the size of `unimath` library almost doubled in the last six months.

## 5   Author Statement

The authors bear all responsibility in case of violation of rights related to the source data, i.e., publicly and freely available libraries in Agda and Lean. The authors also bear all the responsibility associated with the eventual breach of the licenses of the data sources.

---

[2]https://creativecommons.org/licenses/by/4.0/