# OpenReview forum: "MLFMF: Data Sets for Machine Learning for Mathematical Formalization"
_NeurIPS.cc/2023/Track/Datasets_and_Benchmarks — NeurIPS 2023 Datasets and Benchmarks Poster_

### Official Review · Reviewer_XBPs · 2023-06-29
**Interesting datasets, problem with clarity, reproducibility.**

**Rating:** 5
**Confidence:** 4
**Correctness:** NA

**Strengths:**

The motivation of the dataset was clearly explained in the paper, especially in section 2.
This is a challenging application of machine learning, the release of the dataset could help benchmarking machine learning methods in this research field.



**Additional Feedback:**

NA

**Clarity:**


Readers who are unfamiliar with the topics may find it difficult to comprehend the paper. Until section 3.1, there is a lack of a precise definition for a proof assistant, and it would be beneficial to provide an example with a specific entry in the dataset to aid understanding.

The definitions of the data were presented in sections 3.1 and 3.2. However, I found these definitions to be overly abstract. Even after reading these subsections, I still struggle to visualize how computational graphs and the multigraph are structured, especially in relation to a specific example.

Section 3.5 introduces the notation M(u, v) without providing a specific definition. As a result, it is unclear how these notations are connected to the concept of recommendation.

Insufficient attention is given to the sampling of negative edges for link prediction tasks. This omission hinders the ability of readers to replicate the experimental outcomes documented in the paper.




**Documentation:**


The documentation provided for the dataset is inadequate. Despite accessing the anonymous shared Google Drive folder, I was unable to locate useful information regarding the usage of the dataset. It is particularly unclear how to reproduce the results mentioned in the paper using this dataset.


**Limitations:**


The text is challenging to comprehend, requiring substantial improvements before it can be considered for publication. For detailed information on the specific areas that require clarity enhancements, please refer to the Clarity section of my reviews.

In subsection 3.5, examples of machine learning tasks were presented, including link prediction, recommendation, and node classification. I find it challenging to grasp the relationship between solving these tasks and the relevance of recommending mathematical proofs. More specifically, each machine learning task corresponds to a specific step within a mathematical formalization language that is understood by machines. However, it remains unclear how this type of language can be translated into a format that is easily comprehensible to humans?

The documentation accompanying the paper is lacking, as it primarily emphasizes dataset collection while devoting minimal attention to constructing comprehensive documentation. Although some benchmark results are presented, reproducing these results proves to be exceedingly challenging. Furthermore, there is a dearth of specific details regarding the construction of the link prediction model, the hyperparameters employed, and the methodology for sampling negative links.



**Opportunities For Improvement:**


Opportunities For Improvement

I have a suggestion for improving text presentation focuses on providing clearer and more accessible explanations for readers who may not be familiar with the topics discussed. For instance, including an example at the beginning of the paper, specifically in the introduction section, to help these readers better understand the content. Please refer to more comments on the Clarity section of the reviews.

Documentation could be improved, especially the details about how the benchmark models were created, how to execute the codes provided in supplementary materials to get the results reported in the paper for example.

**Relation To Prior Work:**


I am not aware of prior work concerning the same topics that has been officially published. I found this related work https://leandojo.org/ available on arxiv recently, maybe worth mentioning in the related work.



**Summary And Contributions:**

Summary and contributions
A proof assistant enables the algorithmic verification of the accuracy of mathematical proofs and constructions. The papers present a dataset designed for evaluating recommendation systems that aid in the formalization of mathematics using proof assistants.

According to the authors, this dataset contains over 250,000 entries (collected from 4 mathematical formalization library including Agda stdlib,
Agda Unimath, Agda Type Topology and Lean Mathlib4), making it the most extensive collection of formalized mathematical knowledge available in a machine-readable format. I find this dataset interesting because, despite its primary purpose of benchmarking recommendation systems, it has potential for broader applications, such as learning about proof assistants in general.

---

> ### Author Response · Authors · 2023-08-22
> **Response to reviewer XBPs**
>
> Thanks for suggesting an example in the introduction: we added an illustrative example of a simple proof,
> the corresponding part of the network and one of the corresponding s-expressions (DAGs).
>
> We also extended the documentation. The GitHub repository now includes the source code for reproducing the baseline models reported in the paper. We find this revision of the data set repository important for two reasons. First, it extends the documentation and provides readers with a template for learning models from the MLFMF data sets. Second, it assures the reproducibility of the baseline results reported in the paper.
>
> Together, these two improvements should empower ML experts without prior knowledge of proof assistants and formalizing mathematics to evaluate and benchmark their methods for link prediction and machine learning from structured data.
>
> Thanks for providing the link to the Leandojo paper. We noticed that it was published on arXiv weeks after our initial submission.
>
> ### Note
>
> All revised paragraphs are colored in blue.

---

### Official Review · Reviewer_yGPU · 2023-07-20
**Comments**

**Rating:** 5
**Confidence:** 2
**Correctness:** There is no correctness issue.

**Strengths:**

1. The paper is well motivated and the introduction is easy to follow.
2. The authors spend lots of efforts to transform the datasets into different forms so that they can be easily used.

**Additional Feedback:**

No additional feedback.

**Clarity:**

It is a bit challenging to follow the paper without background of mathematical formalization. More real illustrative examples may help.

**Documentation:**

Documentation needs many improvements. There is no readme file and there are too many DAG files to explore.

**Ethics:**

No ethical issues.

**Limitations:**

1. I am not an expert in the field of mathematical formalization. I am not quite clear about the real-world usage of the proposed datasets. What's the real meaning if we conduct link prediction on the given DAG graphs.

2. There are too many small DAG files in the proposed dataset. Are we supposed to merge them into one computational graph? If we want to train an ML model? How do we split the train and test sets (just by files)?

3. The baseline results seem very strange to me. node2vec is almost doing perfect on this data while all other baselines are very close to the Dummy approach. Do we have better choices on these baselines?

**Opportunities For Improvement:**

1. Provide more real illustrative examples for the general ML/DL readers.
2. Provide more readme files on the datasets.
3. Experiment with SOTA baselines on the proposed datasets.

**Relation To Prior Work:**

Previous works on this topic are briefly mentioned in lines 30-37.

**Summary And Contributions:**

In this paper, the authors prepare a new dataset for applying ML approaches in mathematical formalization. The authors conduct preprocessing on these datasets so that different proof assistant can work on these datasets. Results from different baselines are reported and the datasets can be accessible from Google Drive.

---

> ### Author Response · Authors · 2023-08-22
> **Response to reviewer yGPU**
>
> Note that the supplementary materials include essential details about the structure of the data sets. Moreover, we revised and extended the documentation of the data sets. The GitHub repository now includes the source code for reproducing the baseline models reported in the paper. We find this revision of the data set repository important for two reasons. First, it extends the documentation and provides readers with a template for learning models from the MLFMF data sets. Second, it assures the reproducibility of the baseline results reported in the paper.
>
> We also included an illustrative example, which should clarify the relation between the link prediction task and the formalization of mathematics.
>
> We also included the other performance measures in the supplementary materials (Section 2.4.2).
> It is now more evident that node2vec is not doing "almost perfect" as a recommender system
> (see, for example, accuracy@k).
>
> ### Note
>
> All revised paragraphs are colored in blue.

---

### Official Review · Reviewer_zApR · 2023-07-21
**Review for MLFMF Paper**

**Rating:** 6
**Confidence:** 3

**Strengths:**

The authors contribute a relatively large data set suite of mathematical proof data and have made it publicly available to the research community. They appear to have meticulously unified the different library formats to be compatible with one another. They also go into considerable detail about how the data is constructed that would be of interest to subject matter experts.

Assuming that this data set and benchmark is correlated with practical application from an evaluation metric standpoint, it can serve as an important testing bed for new methods to be developed.

**Additional Feedback:**

[Note: Post-rebuttal I have updated my score from 4 to 6 to reflect the improvements made by the authors in the rebuttal]

Minor typo/grammar fixes:

L59 - recomendation -> recommendation
L121 - to learning how do do -> to learning how to do
L168 - every is converted
L225 - Them -> The

One comment I have is that the topics of mathematical proof formulation and proof assistants are not ones I am very familiar with, and so I am open to increasing my score if other reviews provide strong arguments for why certain contributions are significant in those areas.

**Clarity:**

While the paper is structurally correct, it is extremely dense and hard to follow in sections 2, 3.1, 3.2, 3.3, 3.4, and 4.1. The descriptions are very technical and domain-specialized. I felt at many points that it would be infeasible to properly understand what was being communicated without substantial subject matter expertise. For example, Lines 246-247 and Lines 253-254 contain terms like "almost connected", "almost acyclic", "almost DAG-structure" that I was not familiar with. In addition, the abstract itself is very dense and information heavy. I wonder if much of the nuanced descriptions would be better suited to the appendix.

**Correctness:**

The claims and dataset appear to be correct, however due to the complexity of the dataset construction and train-test split, it is very possible that I overlooked a flaw in the approach and/or execution.

The evaluation methods and experiment design appear to be performed correctly, but the results suggest that most methods perform very poorly, while one method performs very strongly, which could be the result of an implementation error or data leak.

**Documentation:**

The documentation appears sufficient for the datasets. For benchmarks, it appears mostly reproducible, however I may have missed key gaps in my reading due to the complexity of the setup of the train-test split. For some methods such as node2vec, hyperparameters of models did not appear to be disclosed, but I did not dive into the code directly.

**Ethics:**

I do not see any ethical concerns with this submission.

**Limitations:**

I do not see expect any negative societal impact from this work. Refer to the prior section for the limitations within the current paper.

**Opportunities For Improvement:**

The experiments appear to lack multiple meaningful baselines. It appears as if all methods except node2vec perform extremely poorly (If I'm not mistaken, they all perform at virtually random guess performance in link prediction). For the one method that performs well, it performs very strongly and thus I question whether the dataset is too easy to base further improvements on.

Little information is provided on the specifics of the node2vec implementation. For example, the usage of the bagged tree model does not include the hyperparameters of the tree, nor the reasoning for the hyperparameters. It was not explained why a bagged tree model was used specifically. Why not a neural network or gradient boosting model?

Why is Lean not used in any experiments, despite it making up ~70% of the data set and the inclusion of multiple proof assistants being a core part of the claimed contribution?

`minRank` seems to be a strange metric to use as a mean. Wouldn't it be better to have a metric such as `top-k accuracy`, with k set to a value such as 5? This would be far more useful when discussing the practical usability as an assistant to a human for recommendations.

"For link prediction, one can use standard classification metrics, such as accuracy, precision, recall, and F1-score. If the model returns its confidence M(u,v) ∈ [0,1] instead of the class value (M (u, v) ∈ {0, 1}, one could additionally consider area under the receiver-operating-characteristic curve. Similarly goes for the recommendation models: one can use precision and recall."

- There is not a justification or explanation for why accuracy was chosen specifically, and it is unclear why the other metrics are discussed in detail when they are not actually used. This further raises the question as to why the other metric scores were not reported.

The benchmark lacks insight into the quality of the dataset itself and justifying how it is an improvement over pre-existing datasets. For example, no ablation on dataset size was performed to show the impact of the "more than 250,000 mathematical formalization entries". It also lacks multiple random seeds.

The benchmark itself uses a very modest amount of compute, running on a single laptop with an 8 year old dual-core CPU. I wonder if the methods tested were limited due to this lack of compute. Would the authors have tested additional baselines given more compute, and if so, which ones?

**Relation To Prior Work:**

The relation to prior work was somewhat discussed in the introduction (L42-L56), but it was challenging to understand what specific differences the given contribution has compared to the prior data sets except its larger size and use of more than one proof assistant. The authors may want to motivate and justify the need for a larger corpus by performing ablation studies that showcase the impact of the size of the training data to the end performance on the benchmark.

**Summary And Contributions:**

The authors propose a set of datasets of mathematical proofs for recommendation system benchmarking. This dataset suite is a combination of multiple mathematical proof libraries, and the authors developed specialized logic to convert the different libraries to a unified format. The correctness of recommendations is evaluated by an automated proof assistant. This suite, with >250,000 total entries, is the largest collection of such mathematical proof data yet available. Through representing the data in a directed multi-graph, the authors propose a method to split the data into train and test for the purposes of evaluation via iterative pruning to generate positive and negative samples. The authors conduct a benchmark comparison on the dataset suite with a variety of baseline methods, showing that node2vec embeddings inputted into a tree-bagging model far outperformed the other baselines.

---

> ### Author Response · Authors · 2023-08-22
> **Respons to reviewer zApR**
>
> ### The performance of the baseline methods
>
> We extended Table 3 with the results of the baseline methods on the Lean library. To this end, we extended both the computational time (to one week) and memory (to 64 GB) available for performing an experiment to be included in the baseline methods report (check the revised paragraphs in Section 4.4).
>
> Bagging is a robust predictive model performing well on various predictive tasks with standard/ recommended hyperparameters settings: a bag of 100 fully grown trees (check the revised paragraphs in Section 4.3). We also opted for this baseline since no elaborative hyperparameter tuning is needed, unlike neural network baselines, which would require numerous design decisions.
>
> At the end of Section 4.4, we added a paragraph commenting on the baseline results and the need for their improvement. Note that supplementary materials now contain the results in terms of the other performance measures that are mentioned in the main text (check Section 2.4.2).
> From the tables present there (Table 5, for example, contains accuracy@k, k = 5), it is evident that
> the performance of node2vec is not *too good to be true*, and there is enough room for improvement.
>
>
> ### Additional baselines
>
> Given more compute time, it would be interesting to see the performance of the graph-based neural networks.
>
>
> ### The choice of the metrics to show
>
> We chose accuracy as this is a common metric for balanced data sets. As evident from the other Tables in Section 2.4.2, the results in terms of other metrics are similar (if we interpret the precision and recall tables together).
> We chose minimal rank as it has a very natural interpretation (the number of wrong recommendations that are ranked higher than any of the correct recommendations).
>
> ### Reproducibility
>
> To address the issue of reproducing the experiments (and provide reassurance that the results are not due to an implementation error or data leak) with the baseline methods, we have published the source code for conducting the experiments in a GitHub repository. The revised version of the repository also improves the existing documentation of the data sets with readme files.
>
> We also included Section 2.4.1 in supplementary materials, which contains a more detailed description of the node2vec experiments.
>
> ### Clarity
>
> We added an illustrative example in the Introduction. We hope that this makes the paper clearer.
>
>
> ### Confusing terms
>
> We apologise for the confusion. The terms "almost connected", "almost DAG", etc. were not used as exact definitions. To avoid confusion, we removed the unnecessary discussion on the (weak) connectivity of the multi-graph.
>
>
> ### Note
>
> All revised paragraphs are colored in blue.

---

> > ### Comment · Reviewer_zApR · 2023-08-29
> > **Response to authors**
> >
> > Firstly, thank you for your detailed response and numerous updates to the main paper and the appendix.
> >
> > Thank you for including Lean in the evaluation, this provides an important comparison between the two datasets.
> >
> > Thank you for providing additional details on the node2vec baseline.
> >
> > Thank you for the illustrative example provided in the introduction, this significantly improves the ease of understanding for those not familiar with the topic, and makes the latter sections easier to parse.
> >
> > ## Remaining Concerns
> >
> > [Important] I am confused why Appendix Table 5 differs so drastically from Table 3's accuracy and `minRank` score. I'm surprised that acc@k shows Dummy outperforming node2vec in multiple cases. How is that possible if `minRank` for node2vec is <5? Wouldn't a `minRank` of `5` require an acc@k=5 of at minimum `0.2`? Yet node2vec's acc@k=5 is `0.14` for Agda stdlib, which doesn't seem correct, unless I'm misunderstanding the metrics. I'd expect `acc@k=5` to correlate very strongly with `minRank`. Further, how does Dummy get `0.16` acc@k, yet has a poor minRank of `218`? A whisker plot for `minRank` would be very useful to better understand what is going on and provide more information than simply reporting the mean. I would appreciate clarification from the authors.
> >
> > I still feel that there is a lack of meaningful baselines besides node2vec. Having at least one other competitive baseline would be very useful (aka a baseline that gets better than random guess accuracy in table 3)
> >
> > There is a lack of ablations, specifically around justifying the utility of the datasets relative to prior work (for example, by subsampling the training data to observe how performance of the models degrades with fewer examples to learn from).
> >
> > ## Conclusion
> >
> > The updates from the authors definitely helped strengthen the paper, but several important questions and concerns still remain as mentioned above. Due to the improvements I have increased my score, but cannot yet recommend acceptance with the current concerns.

---

> > > ### Author Response · Authors · 2023-08-30
> > >
> > > Indeed, it is impossible to have minRank < 5 and acc@5 < 0.2. Thanks for noticing that. That's why we double-checked Table 3 in the main text and Table 5 in the supplementary materials and realized that we wrongly copied the acc@k of node2vec to Table 5. The actual value of acc@5 of node2evc on stdlib is 0.29.
> > >
> > > Regarding the weak "correlation" of minRank and acc@k for Dummy models. The acc@k of the dummy model is relatively high since many entries reference "the most referenced entry". However, if an entry does not reference "the most referenced entry", the minimal rank of the correct entry might be very large (e.g., 8000  since there are approximately 16 000 entries in stdlib)) which leads to seemingly too large average value of minRank.

---

> > > > ### Comment · Reviewer_zApR · 2023-08-30
> > > > **Response to authors**
> > > >
> > > > Thank you for checking and updating the acc@5 values. Regarding the correlation, this makes sense, although I still recommend a whisker plot to better communicate this to readers.
> > > >
> > > > Due to the improvements, I am increasing my score to a 6. I would encourage the authors to refer to my other suggestions for future improvements to the paper, most importantly by adding additional strong baselines.
> > > >
> > > > minor comment: The following text is incorrect after the fixes:
> > > > - "Interestingly, node2vec achieves the best accuracy@k only for Agda unimath and TypeTopology. On the other two data sets, the dummy classifier performs better."

---

> > > > > ### Author Response · Authors · 2023-08-31
> > > > >
> > > > > We performed the additional experiments and extended the supplementary materials in two ways. First, we performed an ablation study of the node2vec model on the Agda stdlib. Second, the results are shown as whisker (or, rather, box) plots; they indeed improve the understanding of results. Thank you for all the useful suggestions. Due to the lack of (CPU) time, we have not considered stronger baselines here, but we will certainly consider them in our future work.

---

### Official Review · Reviewer_F1US · 2023-07-25
**Formalization data set with 250,000 entries and 5 baseline methods**

**Rating:** 8
**Confidence:** 5
**Clarity:** The paper is very clearly written.

**Strengths:**

1. The data set is quite substantial: 250,000 entries from two proof assistants. Further, each data set is represented as a heterogeneous network and a list of s-expressions. This will likely support a new wave of research into automated theorem proving.
2. The baseline methods included in the paper are non-trivial: graph and word embeddings, tree ensembles, and instance-based approaches. Table 2 shows the relative efficacy of these methods.
3. The idea of using a DAG representation of the internal type-theoretic format that is not too closely tied to the syntactic sugar of the two proof assistants gives me hope that others can contribute to this effort in the future and tools built using this library will apply to other proof assistants. It should be highlighted that this design decision comes with a penalty, often replacing kilobytes of source code in the vernacular of the proof assistant with gigabytes of internal representations.


**Additional Feedback:**

The figures on Page 5 use a lot of space that could be better used to discuss hosting, maintenance and even documentation.

**Correctness:**

The data set is constructed in a sound manner and the baseline evaluation methods are sound.

**Documentation:**

Yes, the data set is adequately documented.

**Limitations:**

The authors have used the last paragraph of the conclusion section to identify future work. However, a more explicit discussion of the limitations may be more helpful.

**Opportunities For Improvement:**

The train-test split is not obvious to me. Choosing two threshold p_test and p_body, and then completing proofs with randomly placed "so-called holes" may have interesting consequences. For example, we know that random matrices have exciting properties. Do randomly sampled incomplete proof trees actually represent real-world use cases? I doubt it.
However, this is not a concern to me as the train-test split can be evaluated and perhaps improved by others after this data set and code are published.

**Relation To Prior Work:**

There is a good discussion of the prior research.

**Summary And Contributions:**

The paper presents a new benchmark with 250,000 entries from a library of formalized mathematics written in the Agda and Lean proof assistants. The data set is also supported by 5 baseline implementations with non-trivial performance. Overall, the paper is well-written, and the dataset is likely to be very useful.

---

> ### Author Response · Authors · 2023-08-22
> **Response to reviewer F1US**
>
> ### Discussion on limitations
>
> We moved the discussion related to the train-test split to Conclusions (in the newly added second paragraph on limitations of MLFMF), emphasizing the need for information on the temporal evolution of the library development for a more realistic sampling of test examples.
>
> ### Note
>
> All revised paragraphs are colored in blue.

---

### Official Review · Reviewer_7Q4p · 2023-07-28
**An interesting dataset for promoting future research in the space of recommendations for proof assistants.**

**Rating:** 8
**Confidence:** 3
**Correctness:** I did not find any issues with correc…
**Clarity:** Yes, overall easy to follow.

**Strengths:**

S1: The context that the authors tackle is very relevant and it would be useful to have a public benchmark for this task.

S2: The authors apply their domain-specific knowledge about theorem provers to nicely interpret the original problem into a graph edge prediction problem.

S3: The paper is well-written and easy to follow, even for someone who does not possess a lot of knowledge of theorem provers.

**Additional Feedback:**

n/a

**Documentation:**

The authors provide specific information about the data organization in the supplementary materials.

**Ethics:**

I was not able to identify any ethics issues with this paper.

**Limitations:**

The authors briefly discuss some limitations in the form of directions for future development.

**Opportunities For Improvement:**

I'm not sure how I would improve the paper. The scope is well-defined and the authors provide a solid contribution within that scope.

**Relation To Prior Work:**

The authors mention several previous contributions in this space and contextualize their work relatively well.

**Summary And Contributions:**

This paper focuses on the setting of premise selection, which is the recommendation of theorems that can be used as a basis for proving some desired statement. This is used by automated theorem-proving systems for coming up with novel proofs. The authors take existing libraries of formalized mathematics (in different proof assistants, the authors focus on Agda and Lean) and they transform library entries (theorems, axioms, definitions) into nodes of a graph and construct edges between them if one references the other. They frame the machine learning problem as either an edge prediction task or a recommendation task (while noting that these tasks are equivalent). The resulting datasets are used to benchmark several baseline methods and are also released for use in future research projects.

---

> ### Author Response · Authors · 2023-08-22
> **Response to reviewer 7Q4P**
>
> Thanks for the supportive feedback, especially the comment *"The paper is well-written and easy to follow, even for someone who does not possess a lot of knowledge of theorem provers."*, which is precisely what we aimed at.
>
> #### Note
>
> All revised paragraphs are colored in blue.

---

### Decision · Program_Chairs · 2023-09-22

**Decision:**

Accept (Poster)

**Comment:**

This is a substantial dataset (250,000 entries) with the potential to accelerate development of proof assistants. It also includes 5 non-trivial baselines, along with code to replicate their results. There were some concerns about documentation, which the authors have worked to address. Overall, this is a well-done solution to an important problem. One reviewer wrote: "I'm not sure how I would improve the paper. The scope is well-defined and the authors provide a solid contribution within that scope." Another wrote: "Overall, the paper is well-written, and the dataset is likely to be very useful."